# META-LEARNING FOR SEMI-SUPERVISED FEW-SHOT CLASSIFICATION

**Mengye Ren**[†⋈], **Eleni Triantafillou**[∗†⋈], **Sachin Ravi**[∗§], **Jake Snell**[†⋈], **Kevin Swersky**[¶],
**Joshua B. Tenenbaum**[♮], **Hugo Larochelle**[¶‡] **& Richard S. Zemel**[†‡⋈]
[†]University of Toronto, [§]Princeton University, [¶]Google Brain, [♮]MIT, [‡]CIFAR, [⋈]Vector Institute
{mren,eleni}@cs.toronto.edu, sachinr@cs.princeton.edu,
jsnell@cs.toronto.edu, kswersky@google.com,
jbt@mit.edu, hugolarochelle@google.com, zemel@cs.toronto.edu

## ABSTRACT

In few-shot classification, we are interested in learning algorithms that train a classifier from only a handful of labeled examples. Recent progress in few-shot classification has featured meta-learning, in which a parameterized model for a learning algorithm is defined and trained on episodes representing different classification problems, each with a small labeled training set and its corresponding test set. In this work, we advance this few-shot classification paradigm towards a scenario where unlabeled examples are also available within each episode. We consider two situations: one where all unlabeled examples are assumed to belong to the same set of classes as the labeled examples of the episode, as well as the more challenging situation where examples from other *distractor* classes are also provided. To address this paradigm, we propose novel extensions of Prototypical Networks (Snell et al., 2017) that are augmented with the ability to use unlabeled examples when producing prototypes. These models are trained in an end-to-end way on episodes, to learn to leverage the unlabeled examples successfully. We evaluate these methods on versions of the Omniglot and *mini*ImageNet benchmarks, adapted to this new framework augmented with unlabeled examples. We also propose a new split of ImageNet, consisting of a large set of classes, with a hierarchical structure. Our experiments confirm that our Prototypical Networks can learn to improve their predictions due to unlabeled examples, much like a semi-supervised algorithm would.

## 1 INTRODUCTION

The availability of large quantities of labeled data has enabled deep learning methods to achieve impressive breakthroughs in several tasks related to artificial intelligence, such as speech recognition, object recognition and machine translation. However, current deep learning approaches struggle in tackling problems for which labeled data are scarce. Specifically, while current methods excel at tackling a single problem with lots of labeled data, methods that can simultaneously solve a large variety of problems that each have only a few labels are lacking. Humans on the other hand are readily able to rapidly learn new classes, such as new types of fruit when we visit a tropical country. This significant gap between human and machine learning provides fertile ground for deep learning developments.

For this reason, recently there has been an increasing body of work on few-shot learning, which considers the design of learning algorithms that specifically allow for better generalization on problems with small labeled training sets. Here we focus on the case of few-shot classification, where the given classification problem is assumed to contain only a handful of labeled examples per class. One approach to few-shot learning follows a form of meta-learning [1] (Thrun, 1998; Hochreiter et al., 2001), which performs transfer learning from a pool of various classification problems

---

[∗]Equal contribution.

[1]See the following blog post for an overview: http://bair.berkeley.edu/blog/2017/07/18/learning-to-learn/

generated from large quantities of available labeled data, to new classification problems from classes unseen at training time. Meta-learning may take the form of learning a shared metric (Vinyals et al., 2016; Snell et al., 2017), a common initialization for few-shot classifiers (Ravi & Larochelle, 2017; Finn et al., 2017) or a generic inference network (Santoro et al., 2016; Mishra et al., 2017).

These various meta-learning formulations have led to significant progress recently in few-shot classification. However, this progress has been limited in the setup of each few-shot learning episode, which differs from how humans learn new concepts in many dimensions. In this paper we aim to generalize the setup in two ways. First, we consider a scenario where the new classes are learned in the presence of additional unlabeled data. While there have been many successful applications of semi-supervised learning to the regular setting of a single classification task (Chapelle et al., 2010) where classes at training and test time are the same, such work has not addressed the challenge of performing transfer to new classes never seen at training time, which we consider here. Second, we consider the situation where the new classes to be learned are not viewed in isolation. Instead, many of the unlabeled examples are from different classes; the presence of such *distractor* classes introduces an additional and more realistic level of difficulty to the few-shot problem.

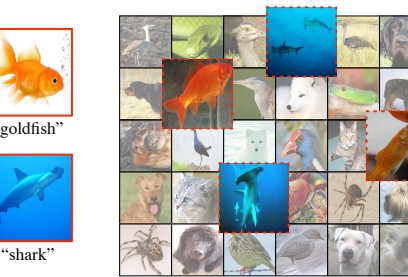

**Figure 1:** Consider a setup where the aim is to learn a classifier to distinguish between two previously unseen classes, goldfish and shark, given not only labeled examples of these two classes, but also a larger pool of unlabeled examples, some of which may belong to one of these two classes of interest. In this work we aim to move a step closer to this more natural learning framework by incorporating in our learning episodes unlabeled data from the classes we aim to learn representations for (shown with dashed red borders) as well as from *distractor* classes .

This work is a first study of this challenging semi-supervised form of few-shot learning. First, we define the problem and propose benchmarks for evaluation that are adapted from the Omniglot and *mini*ImageNet benchmarks used in ordinary few-shot learning. We perform an extensive empirical investigation of the two settings mentioned above, with and without distractor classes. Second, we propose and study three novel extensions of Prototypical Networks (Snell et al., 2017), a state-of-the-art approach to few-shot learning, to the semi-supervised setting. Finally, we demonstrate in our experiments that our semi-supervised variants successfully learn to leverage unlabeled examples and outperform purely supervised Prototypical Networks.

## 2 BACKGROUND

We start by defining precisely the current paradigm for few-shot learning and the Prototypical Network approach to this problem.

### 2.1 FEW-SHOT LEARNING

Recent progress on few-shot learning has been made possible by following an episodic paradigm. Consider a situation where we have a large labeled dataset for a set of classes $\mathcal{C}_{\text{train}}$. However, after training on examples from $\mathcal{C}_{\text{train}}$, our ultimate goal is to produce classifiers for a disjoint set of new classes $\mathcal{C}_{\text{test}}$, for which only a few labeled examples will be available. The idea behind the episodic paradigm is to simulate the types of few-shot problems that will be encountered at test, taking advantage of the large quantities of available labeled data for classes $\mathcal{C}_{\text{train}}$.

Specifically, models are trained on $K$-shot, $N$-way episodes constructed by first sampling a small subset of $N$ classes from $\mathcal{C}_{\text{train}}$ and then generating: 1) a training (support) set $\mathcal{S} = \{(\boldsymbol{x}_1, y_1), (\boldsymbol{x}_2, y_2), \ldots, (\boldsymbol{x}_{N \times K}, y_{N \times K})\}$ containing $K$ examples from each of the $N$ classes and 2) a test (query) set $\mathcal{Q} = \{(\boldsymbol{x}_1^*, y_1^*), (\boldsymbol{x}_2^*, y_2^*), \ldots, (\boldsymbol{x}_T^*, y_T^*)\}$ of different examples from the same $N$ classes. Each $\boldsymbol{x}_i \in \mathbb{R}^D$ is an input vector of dimension $D$ and $y_i \in \{1, 2, \ldots, N\}$ is a class label (similarly for $\boldsymbol{x}_i^*$ and $y_i^*$). Training on such episodes is done by feeding the support set $\mathcal{S}$ to

the model and updating its parameters to minimize the loss of its predictions for the examples in the query set $\mathcal{Q}$.

One way to think of this approach is that our model effectively trains to be a good learning algorithm. Indeed, much like a learning algorithm, the model must take in a set of labeled examples and produce a predictor that can be applied to new examples. Moreover, training directly encourages the classifier produced by the model to have good generalization on the new examples of the query set. Due to this analogy, training under this paradigm is often referred to as learning to learn or meta-learning.

On the other hand, referring to the content of episodes as training and test sets and to the process of learning on these episodes as meta-learning or meta-training (as is sometimes done in the literature) can be confusing. So for the sake of clarity, we will refer to the content of episodes as support and query sets, and to the process of iterating over the training episodes simply as training.

## 2.2 PROTOTYPICAL NETWORKS

Prototypical Network (Snell et al., 2017) is a few-shot learning model that has the virtue of being simple and yet obtaining state-of-the-art performance. At a high-level, it uses the support set $\mathcal{S}$ to extract a prototype vector from each class, and classifies the inputs in the query set based on their distance to the prototype of each class.

More precisely, Prototypical Networks learn an embedding function $h(\boldsymbol{x})$, parameterized as a neural network, that maps examples into a space where examples from the same class are close and those from different classes are far. All parameters of Prototypical Networks lie in the embedding function.

To compute the prototype $\boldsymbol{p}_c$ of each class $c$, a per-class average of the embedded examples is performed:

$$\boldsymbol{p}_c = \frac{\sum_i h(\boldsymbol{x}_i) z_{i,c}}{\sum_i z_{i,c}}, \quad \text{where} \quad z_{i,c} = \mathbb{1}[y_i = c]. \tag{1}$$

These prototypes define a predictor for the class of any new (query) example $\boldsymbol{x}^*$, which assigns a probability over any class $c$ based on the distances between $\boldsymbol{x}^*$ and each prototype, as follows:

$$p(c|\boldsymbol{x}^*, \{\boldsymbol{p}_c\}) = \frac{\exp(-||h(\boldsymbol{x}^*) - \boldsymbol{p}_c||_2^2)}{\sum_{c'} \exp(-||h(\boldsymbol{x}^*) - \boldsymbol{p}_{c'}||_2^2)} \ . \tag{2}$$

The loss function used to update Prototypical Networks for a given training episode is then simply the average negative log-probability of the correct class assignments, for all query examples:

$$-\frac{1}{T} \sum_i \log p(y_i^*|\boldsymbol{x}_i^*, \{\boldsymbol{p}_c\}) \ . \tag{3}$$

Training proceeds by minimizing the average loss, iterating over training episodes and performing a gradient descent update for each.

Generalization performance is measured on test set episodes, which contain images from classes in $\mathcal{C}_{\text{test}}$ instead of $\mathcal{C}_{\text{train}}$. For each test episode, we use the predictor produced by the Prototypical Network for the provided support set $\mathcal{S}$ to classify each of query input $\boldsymbol{x}^*$ into the most likely class $\hat{y} = \text{argmax}_c \, p(c|\boldsymbol{x}^*, \{\boldsymbol{p}_c\})$.

## 3 SEMI-SUPERVISED FEW-SHOT LEARNING

We now define the semi-supervised setting considered in this work for few-shot learning.

The training set is denoted as a tuple of labeled and unlabeled examples: $(\mathcal{S}, \mathcal{R})$. The labeled portion is the usual support set $\mathcal{S}$ of the few-shot learning literature, containing a list of tuples of inputs and targets. In addition to classic few-shot learning, we introduce an unlabeled set $\mathcal{R}$ containing only inputs: $\mathcal{R} = \{\tilde{\boldsymbol{x}}_1, \tilde{\boldsymbol{x}}_2, \ldots, \tilde{\boldsymbol{x}}_M\}$. As in the purely supervised setting, our models are trained to perform well when predicting the labels for the examples in the episode's query set $\mathcal{Q}$. Figure 2 shows a visualization of training and test episodes.

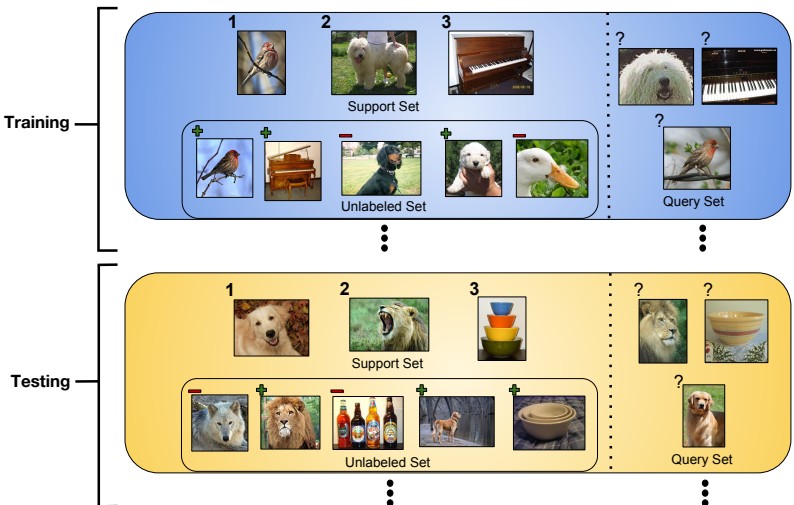

**Figure 2:** Example of the semi-supervised few-shot learning setup. Training involves iterating through training episodes, consisting of a support set $\mathcal{S}$, an unlabeled set $\mathcal{R}$, and a query set $\mathcal{Q}$. The goal is to use the labeled items (shown with their numeric class label) in $\mathcal{S}$ and the unlabeled items in $\mathcal{R}$ within each episode to generalize to good performance on the corresponding query set. The unlabeled items in $\mathcal{R}$ may either be pertinent to the classes we are considering (shown above with green plus signs) or they may be *distractor* items which belong to a class that is not relevant to the current episode (shown with red minus signs). However note that the model does not actually have ground truth information as to whether each unlabeled example is a distractor or not; the plus/minus signs are shown only for illustrative purposes. At test time, we are given new episodes consisting of novel classes not seen during training that we use to evaluate the meta-learning method.

## 3.1 SEMI-SUPERVISED PROTOTYPICAL NETWORKS

In their original formulation, Prototypical Networks do not specify a way to leverage the unlabeled set $\mathcal{R}$. In what follows, we now propose various extensions that start from the basic definition of prototypes $\boldsymbol{p}_c$ and provide a procedure for producing refined prototypes $\tilde{\boldsymbol{p}}_c$ using the unlabeled examples in $\mathcal{R}$.

After the refined prototypes are obtained, each of these models is trained with the same loss function for ordinary Prototypical Networks of Equation 3, but replacing $\boldsymbol{p}_c$ with $\tilde{\boldsymbol{p}}_c$. That is, each query example is classified into one of the $N$ classes based on the proximity of its embedded position with the corresponding *refined* prototypes, and the average negative log-probability of the correct classification is used for training.

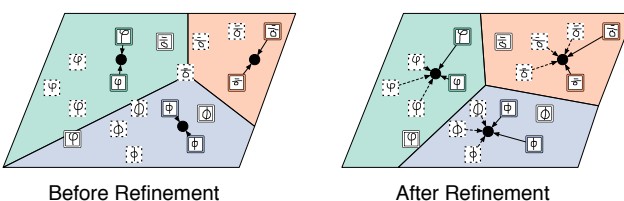

**Figure 3:** Left: The prototypes are initialized based on the mean location of the examples of the corresponding class, as in ordinary Prototypical Networks. Support, unlabeled, and query examples have solid, dashed, and white colored borders respectively. Right: The refined prototypes obtained by incorporating the unlabeled examples, which classifies all query examples correctly.

### 3.1.1 PROTOTYPICAL NETWORKS WITH SOFT $k$-MEANS

We first consider a simple way of leveraging unlabeled examples for refining prototypes, by taking inspiration from semi-supervised clustering. Viewing each prototype as a cluster center, the refinement process could attempt to adjust the cluster locations to better fit the examples in both the support and unlabeled sets. Under this view, cluster assignments of the labeled examples in the support set are considered known and fixed to each example's label. The refinement process must instead estimate the cluster assignments of the unlabeled examples and adjust the cluster locations (the prototypes) accordingly.

One natural choice would be to borrow from the inference performed by soft $k$-means. We prefer this version of $k$-means over hard assignments since hard assignments would make the inference non-differentiable. We start with the regular Prototypical Network's prototypes $\boldsymbol{p}_c$ (as specified in Equation 1) as the cluster locations. Then, the unlabeled examples get a partial assignment ($\tilde{z}_{j,c}$) to each cluster based on their Euclidean distance to the cluster locations. Finally, refined prototypes are obtained by incorporating these unlabeled examples.

This process can be summarized as follows:

$$\tilde{\boldsymbol{p}}_c = \frac{\sum_i h(\boldsymbol{x}_i)z_{i,c} + \sum_j h(\tilde{\boldsymbol{x}}_j)\tilde{z}_{j,c}}{\sum_i z_{i,c} + \sum_j \tilde{z}_{j,c}}, \quad \text{where} \quad \tilde{z}_{j,c} = \frac{\exp\left(-||h(\tilde{\boldsymbol{x}}_j) - \boldsymbol{p}_c||_2^2\right)}{\sum_{c'} \exp\left(-||h(\tilde{\boldsymbol{x}}_j) - \boldsymbol{p}_{c'}||_2^2\right)} \quad (4)$$

Predictions of each query input's class is then modeled as in Equation 2, but using the refined prototypes $\tilde{\boldsymbol{p}}_c$.

We could perform several iterations of refinement, as is usual in $k$-means. However, we have experimented with various number of iterations and found results to not improve beyond a single refinement step.

### 3.1.2 PROTOTYPICAL NETWORKS WITH SOFT $k$-MEANS WITH A DISTRACTOR CLUSTER

The soft $k$-means approach described above implicitly assumes that each unlabeled example belongs to either one of the $N$ classes in the episode. However, it would be much more general to not make that assumption and have a model robust to the existence of examples from other classes, which we refer to as distractor classes. For example, such a situation would arise if we wanted to distinguish between pictures of unicycles and scooters, and decided to add an unlabeled set by downloading images from the web. It then would not be realistic to assume that all these images are of unicycles or scooters. Even with a focused search, some may be from similar classes, such as bicycle.

Since soft $k$-means distributes its soft assignments across all classes, distractor items could be harmful and interfere with the refinement process, as prototypes would be adjusted to also partially account for these distractors. A simple way to address this is to add an additional cluster whose purpose is to capture the distractors, thus preventing them from polluting the clusters of the classes of interest:

$$\boldsymbol{p}_c = \begin{cases} \frac{\sum_i h(\boldsymbol{x}_i)z_{i,c}}{\sum_i z_{i,c}} & \text{for } c = 1...N \\ \boldsymbol{0} & \text{for } c = N+1 \end{cases} \quad (5)$$

Here we take the simplifying assumption that the distractor cluster has a prototype centered at the origin. We also consider introducing length-scales $r_c$ to represent variations in the within-cluster distances, specifically for the distractor cluster:

$$\tilde{z}_{j,c} = \frac{\exp\left(-\frac{1}{r_c^2}||\tilde{\boldsymbol{x}}_j - \boldsymbol{p}_c||_2^2 - A(r_c)\right)}{\sum_{c'} \exp\left(-\frac{1}{r_c^2}||\tilde{\boldsymbol{x}}_j - \boldsymbol{p}_{c'}||_2^2 - A(r_{c'})\right)}, \quad \text{where} \quad A(r) = \frac{1}{2}\log(2\pi) + \log(r) \quad (6)$$

For simplicity, we set $r_{1...N}$ to 1 in our experiments, and only learn the length-scale of the distractor cluster $r_{N+1}$.

### 3.1.3 PROTOTYPICAL NETWORKS WITH SOFT $k$-MEANS AND MASKING

Modeling distractor unlabeled examples with a single cluster is likely too simplistic. Indeed, it is inconsistent with our assumption that each cluster corresponds to one class, since distractor examples may very well cover more than a single natural object category. Continuing with our unicycles and bicycles example, our web search for unlabeled images could accidentally include not only bicycles, but other related objects such as tricycles or cars. This was also reflected in our experiments, where we constructed the episode generating process so that it would sample distractor examples from multiple classes.

To address this problem, we propose an improved variant: instead of capturing distractors with a high-variance catch-all cluster, we model distractors as examples that are not within some area of any of the legitimate class prototypes. This is done by incorporating a soft-masking mechanism on

the contribution of unlabeled examples. At a high level, we want unlabeled examples that are closer to a prototype to be masked less than those that are farther.

More specifically, we modify the soft $k$-means refinement as follows. We start by computing normalized distances $\tilde{d}_{j,c}$ between examples $\tilde{\boldsymbol{x}}_j$ and prototypes $\boldsymbol{p}_c$:

$$\tilde{d}_{j,c} = \frac{d_{j,c}}{\frac{1}{M}\sum_j d_{j,c}}, \text{ where } d_{j,c} = ||h(\tilde{\boldsymbol{x}}_j) - \boldsymbol{p}_c||_2^2 \tag{7}$$

Then, soft thresholds $\beta_c$ and slopes $\gamma_c$ are predicted for each prototype, by feeding to a small neural network various statistics of the normalized distances for the prototype:

$$[\beta_c, \gamma_c] = \text{MLP}\left(\left[\min_j(\tilde{d}_{j,c}), \max_j(\tilde{d}_{j,c}), \text{var}_j(\tilde{d}_{j,c}), \text{skew}_j(\tilde{d}_{j,c}), \text{kurt}_j(\tilde{d}_{j,c})\right]\right) \tag{8}$$

This allows each threshold to use information on the amount of intra-cluster variation to determine how aggressively it should cut out unlabeled examples.

Then, soft masks $m_{j,c}$ for the contribution of each example to each prototype are computed, by comparing to the threshold the normalized distances, as follows:

$$\tilde{\boldsymbol{p}}_c = \frac{\sum_i h(\boldsymbol{x}_i)z_{i,c} + \sum_j h(\tilde{\boldsymbol{x}}_j)\tilde{z}_{j,c}m_{j,c}}{\sum_i z_{i,c} + \sum_j \tilde{z}_{j,c}m_{j,c}}, \text{ where } m_{j,c} = \sigma\left(-\gamma_c\left(\tilde{d}_{j,c} - \beta_c\right)\right) \tag{9}$$

where $\sigma(\cdot)$ is the sigmoid function.

When training with this refinement process, the model can now use its MLP in Equation 8 to learn to include or ignore entirely certain unlabeled examples. The use of soft masks makes this process entirely differentiable[2]. Finally, much like for regular soft $k$-means (with or without a distractor cluster), while we could recursively repeat the refinement for multiple steps, we found a single step to perform well enough.

## 4  RELATED WORK

We summarize here the most relevant work from the literature on few-shot learning, semi-supervised learning and clustering.

The best performing methods for few-shot learning use the episodic training framework prescribed by meta-learning. The approach within which our work falls is that of metric learning methods. Previous work in metric-learning for few-shot-classification includes Deep Siamese Networks (Koch et al., 2015), Matching Networks (Vinyals et al., 2016), and Prototypical Networks (Snell et al., 2017), which is the model we extend to the semi-supervised setting in our work. The general idea here is to learn an embedding function that embeds examples belonging to the same class close together while keeping embeddings from separate classes far apart. Distances between embeddings of items from the support set and query set are then used as a notion of similarity to do classification. Lastly, closely related to our work with regard to extending the few-shot learning setting, Bachman et al. (2017) employ Matching Networks in an active learning framework where the model has a choice of which unlabeled item to add to the support set over a certain number of time steps before classifying the query set. Unlike our setting, their meta-learning agent can acquire ground-truth labels from the unlabeled set, and they do not use distractor examples.

Other meta-learning approaches to few-shot learning include learning how to use the support set to update a learner model so as to generalize to the query set. Recent work has involved learning either the weight initialization and/or update step that is used by a learner neural network (Ravi & Larochelle, 2017; Finn et al., 2017). Another approach is to train a generic neural architecture such as a memory-augmented recurrent network (Santoro et al., 2016) or a temporal convolutional network (Mishra et al., 2017) to sequentially process the support set and perform accurate predictions of the labels of the query set examples. These other methods are also competitive for few-shot learning, but we chose to extend Prototypical Networks in this work for its simplicity and efficiency.

---

[2]We stop gradients from passing through the computation of the statistics in Equation 8, to avoid potential numerical instabilities.

As for the literature on semi-supervised learning, while it is quite vast (Zhu, 2005; Chapelle et al., 2010), the most relevant category to our work is related to self-training (Yarowsky, 1995; Rosenberg et al., 2005). Here, a classifier is first trained on the initial training set. The classifier is then used to classify unlabeled items, and the most confidently predicted unlabeled items are added to the training set with the prediction of the classifier as the assumed label. This is similar to our soft $k$-Means extension to Prototypical Networks. Indeed, since the soft assignments (Equation 4) match the regular Prototypical Network's classifier output for new inputs (Equation 2), then the refinement can be thought of re-feeding to a Prototypical Network a new support set augmented with (soft) self-labels from the unlabeled set.

Our algorithm is also related to transductive learning (Vapnik, 1998; Joachims, 1999; Fu et al., 2015), where the base classifier gets refined by seeing the unlabeled examples. In practice, one could use our method in a transductive setting where the unlabeled set is the same as the query set; however, here to avoid our model memorizing labels of the unlabeled set during the meta-learning procedure, we split out a separate unlabeled set that is different from the query set.

In addition to the original $k$-Means method (Lloyd, 1982), the most related work to our setup involving clustering algorithms considers applying $k$-Means in the presence of outliers (Hautamäki et al., 2005; Chawla & Gionis, 2013; Gupta et al., 2017). The goal here is to correctly discover and ignore the outliers so that they do not wrongly shift the cluster locations to form a bad partition of the true data. This objective is also important in our setup as not ignoring outliers (or distractors) will wrongly shift the prototypes and negatively influence classification performance.

Our contribution to the semi-supervised learning and clustering literature is to go beyond the classical setting of training and evaluating within a single dataset, and consider the setting where we must learn to transfer from a set of training classes $\mathcal{C}_{\text{train}}$ to a new set of test classes $\mathcal{C}_{\text{test}}$.

## 5 EXPERIMENTS

### 5.1 DATASETS

We evaluate the performance of our model on three datasets: two benchmark few-shot classification datasets and a novel large-scale dataset that we hope will be useful for future few-shot learning work.

**Omniglot** (Lake et al., 2011) is a dataset of 1,623 handwritten characters from 50 alphabets. Each character was drawn by 20 human subjects. We follow the few-shot setting proposed by Vinyals et al. (2016), in which the images are resized to $28 \times 28$ pixels and rotations in multiples of $90°$ are applied, yielding 6,492 classes in total. These are split into 4,112 training classes, 688 validation classes, and 1,692 testing classes.

*mini*ImageNet (Vinyals et al., 2016) is a modified version of the ILSVRC-12 dataset (Russakovsky et al., 2015), in which 600 images for each of 100 classes were randomly chosen to be part of the dataset. We rely on the class split used by Ravi & Larochelle (2017). These splits use 64 classes for training, 16 for validation, and 20 for test. All images are of size $84 \times 84$ pixels.

*tiered*ImageNet is our proposed dataset for few-shot classification. Like *mini*Imagenet, it is a subset of ILSVRC-12. However, *tiered*ImageNet represents a larger subset of ILSVRC-12 (608 classes rather than 100 for *mini*ImageNet). Analogous to Omniglot, in which characters are grouped into alphabets, *tiered*ImageNet groups classes into broader categories corresponding to higher-level nodes in the ImageNet (Deng et al., 2009) hierarchy. There are 34 categories in total, with each category containing between 10 and 30 classes. These are split into 20 training, 6 validation and 8 testing categories (details of the dataset can be found in the supplementary material). This ensures that all of the training classes are sufficiently distinct from the testing classes, unlike *mini*ImageNet and other alternatives such as *rand*ImageNet proposed by Vinyals et al. (2016). For example, "pipe organ" is a training class and "electric guitar" is a test class in the Ravi & Larochelle (2017) split of *mini*Imagenet, even though they are both musical instruments. This scenario would not occur in *tiered*ImageNet since "musical instrument" is a high-level category and as such is not split between training and test classes. This represents a more realistic few-shot learning scenario since in general we cannot assume that test classes will be similar to those seen in training. Additionally, the tiered structure of *tiered*ImageNet may be useful for few-shot learning approaches that can take advantage of hierarchical relationships between classes. We leave such interesting extensions for future work.

| Models | Acc. | Acc. w/ D |
|---|---|---|
| Supervised | $94.62 \pm 0.09$ | $94.62 \pm 0.09$ |
| Semi-Supervised Inference | $97.45 \pm 0.05$ | $95.08 \pm 0.09$ |
| Soft $k$-Means | $97.25 \pm 0.10$ | $95.01 \pm 0.09$ |
| Soft $k$-Means+Cluster | $\mathbf{97.68 \pm 0.07}$ | $97.17 \pm 0.04$ |
| Masked Soft $k$-Means | $97.52 \pm 0.07$ | $\mathbf{97.30 \pm 0.08}$ |

**Table 1:** Omniglot 1-shot classification results. In this table as well as those below "w/ D" denotes "with distractors", where the unlabeled images contain irrelevant classes.

## 5.2 ADAPTING THE DATASETS FOR SEMI-SUPERVISED LEARNING

For each dataset, we first create an additional split to separate the images of each class into disjoint labeled and unlabeled sets. For Omniglot and *tiered*ImageNet we sample 10% of the images of each class to form the labeled split. The remaining 90% can only be used in the unlabeled portion of episodes. For *mini*ImageNet we use 40% of the data for the labeled split and the remaining 60% for the unlabeled, since we noticed that 10% was too small to achieve reasonable performance and avoid overfitting. We report the average classification scores over 10 random splits of labeled and unlabeled portions of the training set, with uncertainty computed in standard error (standard deviation divided by the square root of the total number of splits).

We would like to emphasize that due to this labeled/unlabeled split, we are using strictly less label information than in the previously-published work on these datasets. Because of this, we do not expect our results to match the published numbers, which should instead be interpreted as an upper-bound for the performance of the semi-supervised models defined in this work.

Episode construction then is performed as follows. For a given dataset, we create a training episode by first sampling $N$ classes uniformly at random from the set of training classes $\mathcal{C}_{\mathrm{train}}$. We then sample $K$ images from the labeled split of each of these classes to form the support set, and $M$ images from the unlabeled split of each of these classes to form the unlabeled set. Optionally, when including distractors, we additionally sample $H$ other classes from the set of training classes and $M$ images from the unlabeled split of each to act as the distractors. These distractor images are added to the unlabeled set along with the unlabeled images of the $N$ classes of interest (for a total of $MN + MH$ unlabeled images). The query portion of the episode is comprised of a fixed number of images from the labeled split of each of the $N$ chosen classes. Test episodes are created analogously, but with the $N$ classes (and optionally the $H$ distractor classes) sampled from $\mathcal{C}_{\mathrm{test}}$. In the experiments reported here we used $H = N = 5$, i.e. 5 classes for both the labeled classes and the distractor classes. We used $M = 5$ for training and $M = 20$ for testing in most cases, thus measuring the ability of the models to generalize to a larger unlabeled set size. Details of the dataset splits, including the specific classes assigned to train/validation/test sets, can be found in Appendices A and B.

In each dataset we compare our three semi-supervised models with two baselines. The first baseline, referred to as "Supervised" in our tables, is an ordinary Prototypical Network that is trained in a purely supervised way on the labeled split of each dataset. The second baseline, referred to as "Semi-Supervised Inference", uses the embedding function learned by this supervised Prototypical Network, but performs semi-supervised refinement of the prototypes at test time using a step of Soft $k$-Means refinement. This is to be contrasted with our semi-supervised models that perform this refinement both at training time and at test time, therefore learning a different embedding function. We evaluate each model in two settings: one where all unlabeled examples belong to the classes of interest, and a more challenging one that includes distractors. Details of the model hyperparameters can be found in Appendix D and our online repository.[3]

## 5.3 RESULTS

Results for Omniglot, *mini*ImageNet and *tiered*ImageNet are given in Tables 1, 2 and 5, respectively, while Figure 4 shows the performance of our models on *tiered*ImageNet (our largest dataset) using

---

[3] Code available at `https://github.com/renmengye/few-shot-ssl-public`

| Models | 1-shot Acc. | 5-shot Acc. | 1-shot Acc w/ D | 5-shot Acc. w/ D |
|---|---|---|---|---|
| Supervised | $43.61 \pm 0.27$ | $59.08 \pm 0.22$ | $43.61 \pm 0.27$ | $59.08 \pm 0.22$ |
| Semi-Supervised Inference | $48.98 \pm 0.34$ | $63.77 \pm 0.20$ | $47.42 \pm 0.33$ | $62.62 \pm 0.24$ |
| Soft $k$-Means | $\mathbf{50.09 \pm 0.45}$ | $\mathbf{64.59 \pm 0.28}$ | $\mathbf{48.70 \pm 0.32}$ | $\mathbf{63.55 \pm 0.28}$ |
| Soft $k$-Means+Cluster | $49.03 \pm 0.24$ | $63.08 \pm 0.18$ | $\mathbf{48.86 \pm 0.32}$ | $61.27 \pm 0.24$ |
| Masked Soft $k$-Means | $\mathbf{50.41 \pm 0.31}$ | $\mathbf{64.39 \pm 0.24}$ | $\mathbf{49.04 \pm 0.31}$ | $62.96 \pm 0.14$ |

**Table 2:** *mini*ImageNet 1/5-shot classification results.

| Models | 1-shot Acc. | 5-shot Acc. | 1-shot Acc. w/ D | 5-shot Acc. w/ D |
|---|---|---|---|---|
| Supervised | $46.52 \pm 0.52$ | $66.15 \pm 0.22$ | $46.52 \pm 0.52$ | $66.15 \pm 0.22$ |
| Semi-Supervised Inference | $50.74 \pm 0.75$ | $69.37 \pm 0.26$ | $48.67 \pm 0.60$ | $67.46 \pm 0.24$ |
| Soft $k$-Means | $51.52 \pm 0.36$ | $\mathbf{70.25 \pm 0.31}$ | $49.88 \pm 0.52$ | $68.32 \pm 0.22$ |
| Soft $k$-Means+Cluster | $\mathbf{51.85 \pm 0.25}$ | $69.42 \pm 0.17$ | $\mathbf{51.36 \pm 0.31}$ | $67.56 \pm 0.10$ |
| Masked Soft $k$-Means | $\mathbf{52.39 \pm 0.44}$ | $69.88 \pm 0.20$ | $\mathbf{51.38 \pm 0.38}$ | $\mathbf{69.08 \pm 0.25}$ |

**Table 3:** *tiered*ImageNet 1/5-shot classification results.

different values for $M$ (number of items in the unlabeled set per class). Additional results comparing the ProtoNet model to various baselines on these datasets, and analysis of the performance of the Masked Soft $k$-Means model can be found in Appendix C.

Across all three benchmarks, at least one of our proposed models outperforms the baselines, demonstrating the effectiveness of our semi-supervised meta-learning procedure. In the non-distractor settings, all three proposed models outperform the baselines in almost all the experiments, without a clear winner between the three models across the datasets and shot numbers. In the scenario where training and testing includes distractors, Masked Soft $k$-Means shows the most robust performance across all three datasets, attaining the best results in each case but one. In fact this model reaches performance that is close to the upper bound based on the results without distractors.

From Figure 4, we observe clear improvements in test accuracy when the number of items in the unlabeled set per class grows from 0 to 25. These models were trained with $M = 5$ and thus are showing an ability to extrapolate in generalization. This confirms that, through meta-training, the models learn to acquire a better representation that is improved by semi-supervised refinement.

## 6 CONCLUSION

In this work, we propose a novel semi-supervised few-shot learning paradigm, where an unlabeled set is added to each episode. We also extend the setup to more realistic situations where the unlabeled set has novel classes distinct from the labeled classes. To address the problem that current few-shot classification datasets are too small for a labeled vs. unlabeled split and also lack hierarchical levels of labels, we introduce a new dataset, *tiered*ImageNet. We propose several novel extensions of Prototypical Networks, and they show consistent improvements under semi-supervised settings compared to our baselines. As future work, we are working on incorporating fast weights (Ba et al., 2016; Finn et al., 2017) into our framework so that examples can have different embedding representations given the contents in the episode.

**Acknowledgement**   Supported by grants from NSERC, Samsung, and the Intelligence Advanced Research Projects Activity (IARPA) via Department of Interior/Interior Business Center (DoI/IBC) contract number D16PC00003. The U.S. Government is authorized to reproduce and distribute reprints for Governmental purposes notwithstanding any copyright annotation thereon. Disclaimer: The views and conclusions contained herein are those of the authors and should not be interpreted as necessarily representing the official policies or endorsements, either expressed or implied, of IARPA, DoI/IBC, or the U.S. Government.

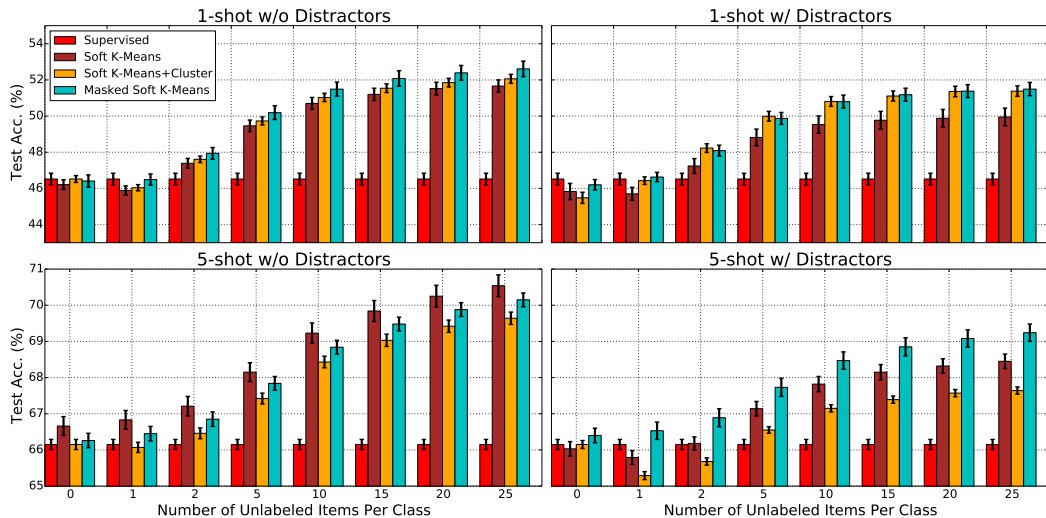

**Figure 4:** Model Performance on *tiered*ImageNet with different numbers of unlabeled items during test time.

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

## A    Omniglot Dataset Details

We used the following split details for experiments on Omniglot dataset. This is the same train/test split as (Vinyals et al., 2016), but we created our own validation split for selecting hyper-parameters. Models are trained on the train split only.

**Train Alphabets:** Alphabet_of_the_Magi, Angelic, Anglo-Saxon_Futhorc, Arcadian, Asomtavruli_(Georgian), Atemayar_Qelisayer, Atlantean, Aurek-Besh, Avesta, Balinese, Blackfoot_(Canadian_Aboriginal_Syllabics), Braille, Burmese_(Myanmar), Cyrillic, Futurama, Ge_ez, Glagolitic, Grantha, Greek, Gujarati, Gurmukhi (character 01-41), Inuktitut_(Canadian_Aboriginal_Syllabics), Japanese_(hiragana), Japanese_(katakana), Korean, Latin, Malay_(Jawi_-_Arabic), N_Ko, Ojibwe_(Canadian_Aboriginal_Syllabics), Sanskrit, Syriac_(Estrangelo), Tagalog, Tifinagh

**Validation Alphabets:** Armenian, Bengali, Early_Aramaic, Hebrew, Mkhedruli_(Geogian)

**Test Alphabets:** Gurmukhi (character 42-45), Kannada, Keble, Malayalam, Manipuri, Mongolian, Old_Church_Slavonic_(Cyrillic), Oriya, Sylheti, Syriac_(Serto), Tengwar, Tibetan, ULOG

## B    *tiered*ImageNet Dataset Details

Each high-level category in *tiered*ImageNet contains between 10 and 30 ILSVRC-12 classes (17.8 on average). In the ImageNet hierarchy, some classes have multiple parent nodes. Therefore, classes belonging to more than one category were removed from the dataset to ensure separation between training and test categories. Test categories were chosen to reflect various levels of separation between training and test classes. Some test categories (such as "working dog") are fairly similar to training categories, whereas others (such as "geological formation") are quite different. The list of categories is shown below and statistics of the dataset can be found in Table 4. A visualization of the categories according to the ImageNet hierarchy is shown in Figure 5. The full list of classes per category will also be made public, however for the sake of brevity we do not include it here.

**Table 4:** Statistics of the *tiered*ImageNet dataset.

|            | Train   | Val     | Test    | Total   |
|------------|---------|---------|---------|---------|
| Categories | 20      | 6       | 8       | 34      |
| Classes    | 351     | 97      | 160     | 608     |
| Images     | 448,695 | 124,261 | 206,209 | 779,165 |

**Train Categories**: n02087551 (hound, hound dog), n02092468 (terrier), n02120997 (feline, felid), n02370806 (ungulate, hoofed mammal), n02469914 (primate), n01726692 (snake, serpent, ophidian), n01674216 (saurian), n01524359 (passerine, passeriform bird), n01844917 (aquatic bird), n04081844 (restraint, constraint), n03574816 (instrument), n03800933 (musical instrument, instrument), n03125870 (craft), n04451818 (tool), n03414162 (game equipment), n03278248 (electronic equipment), n03419014 (garment), n03297735 (establishment), n02913152 (building, edifice), n04014297 (protective covering, protective cover, protection).

**Validation Categories**: n02098550 (sporting dog, gun dog), n03257877 (durables, durable goods, consumer durables), n03405265 (furnishing), n03699975 (machine), n03738472 (mechanism), n03791235 (motor vehicle, automotive vehicle),

**Test Categories**: n02103406 (working dog), n01473806 (aquatic vertebrate), n02159955 (insect), n04531098 (vessel), n03839993 (obstruction, obstructor, obstructer, impediment, impedimenta), n09287968 (geological formation, formation), n00020090 (substance), n15046900 (solid).

| Models | Omniglot | *mini*ImageNet | | *tiered*ImageNet | |
| | 1-shot | 1-shot | 5-shot | 1-shot | 5-shot |
|---|---|---|---|---|---|
| 1-NN Pixel | $40.39 \pm 0.36$ | $26.74 \pm 0.48$ | $31.43 \pm 0.51$ | $26.55 \pm 0.50$ | $30.79 \pm 0.53$ |
| 1-NN CNN rnd | $59.55 \pm 0.46$ | $24.03 \pm 0.38$ | $27.54 \pm 0.42$ | $25.49 \pm 0.45$ | $30.01 \pm 0.47$ |
| 1-NN CNN pre | $52.53 \pm 0.51$ | $32.90 \pm 0.58$ | $40.79 \pm 0.76$ | $32.76 \pm 0.66$ | $40.26 \pm 0.67$ |
| LR Pixel | $49.15 \pm 0.39$ | $24.50 \pm 0.41$ | $33.33 \pm 0.68$ | $25.70 \pm 0.46$ | $36.30 \pm 0.62$ |
| LR CNN rnd | $57.80 \pm 0.45$ | $24.10 \pm 0.50$ | $28.40 \pm 0.42$ | $26.55 \pm 0.48$ | $32.51 \pm 0.52$ |
| LR CNN pre | $48.49 \pm 0.47$ | $30.28 \pm 0.54$ | $40.27 \pm 0.59$ | $34.52 \pm 0.68$ | $43.58 \pm 0.72$ |
| ProtoNet | $\mathbf{94.62 \pm 0.09}$ | $\mathbf{43.61 \pm 0.27}$ | $\mathbf{59.08 \pm 0.22}$ | $\mathbf{46.52 \pm 0.32}$ | $\mathbf{66.15 \pm 0.34}$ |

**Table 5:** Few-shot learning baseline results using labeled/unlabeled splits. Baselines either takes inputs directly from the pixel space or use a CNN to extract features. "rnd" denotes using a randomly initialized CNN, and "pre" denotes using a CNN that is pretrained for supervised classification for all training classes.

## C    EXTRA EXPERIMENTAL RESULTS

### C.1    FEW-SHOT CLASSIFICATION BASELINES

We provide baseline results on few-shot classification using 1-nearest neighbor and logistic regression with either pixel inputs or CNN features. Compared with the baselines, Regular ProtoNet performs significantly better on all three few-shot classification datasets.

### C.2    NUMBER OF UNLABELED ITEMS

Figure 6 shows test accuracy values with different number of unlabeled items during test time. Figure 7 shows our mask output value distribution of the Masked Soft $k$-Means model on Omniglot. The mask values have a bi-modal distribution, corresponding to distractor and non-distractor items.

## D    HYPERPARAMETER DETAILS

For Omniglot, we adopted the best hyperparameter settings found for ordinary Prototypical Networks in Snell et al. (2017). In these settings, the learning rate was set to 1e-3, and cut in half every 2K updates starting at update 2K. We trained for a total of 20K updates. For *mini*Imagenet and *tiered*ImageNet, we trained with a starting learning rate of 1e-3, which we also decayed. We started the decay after 25K updates, and every 25K updates thereafter we cut it in half. We trained for a total of 200K updates. We used ADAM (Kingma & Ba, 2014) for the optimization of our models. For the MLP used in the Masked Soft $k$-Means model, we use a single hidden layer with 20 hidden units with a tanh non-linearity for all 3 datasets. We did not tune the hyparameters of this MLP so better performance may be attained with a more rigorous hyperparameter search.

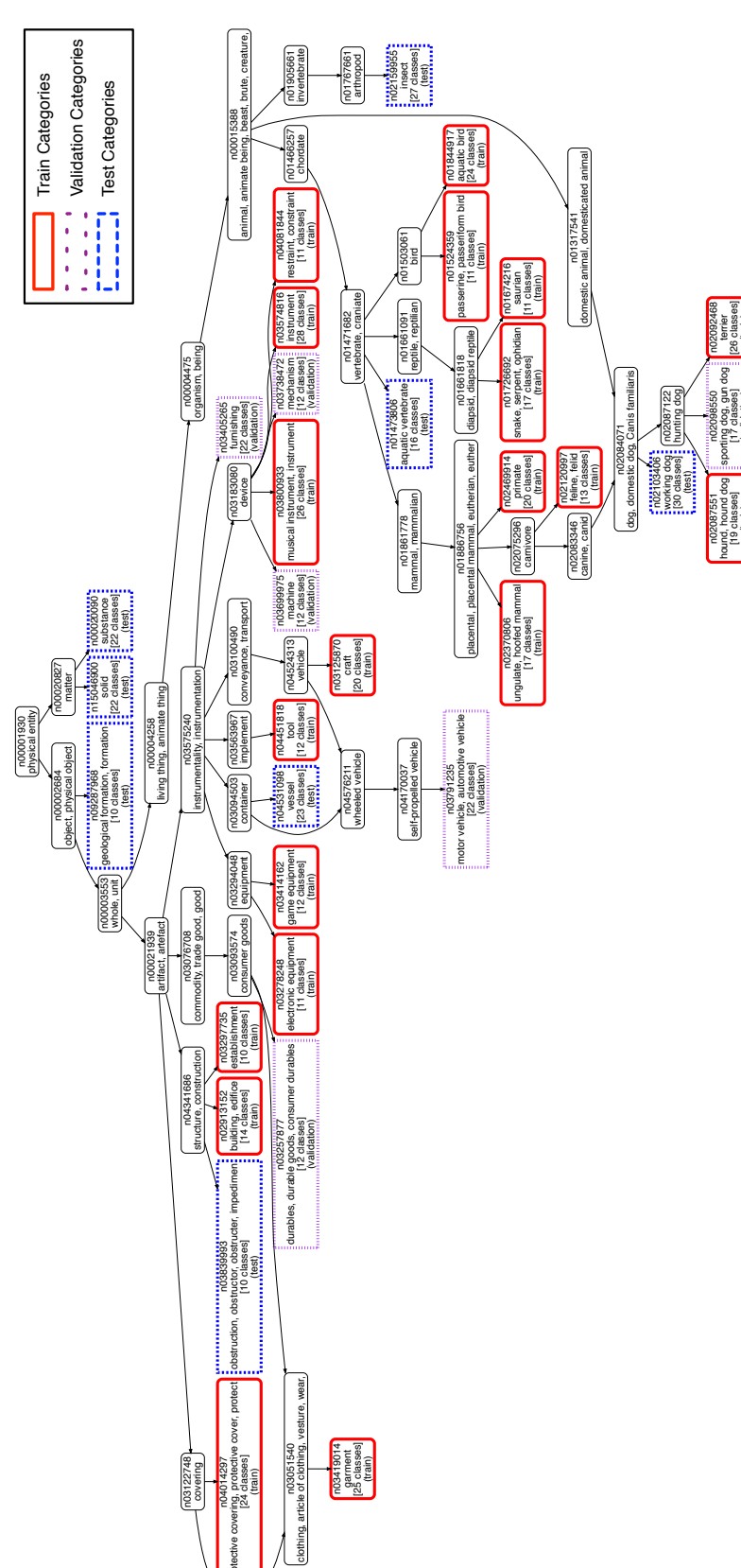

**Figure 5:** Hierarchy of *tiered*Imagenet categories. Training categories are highlighted in red and test categories in blue. Each category indicates the number of associated classes from ILSVRC-12. Best viewed zoomed-in on electronic version.

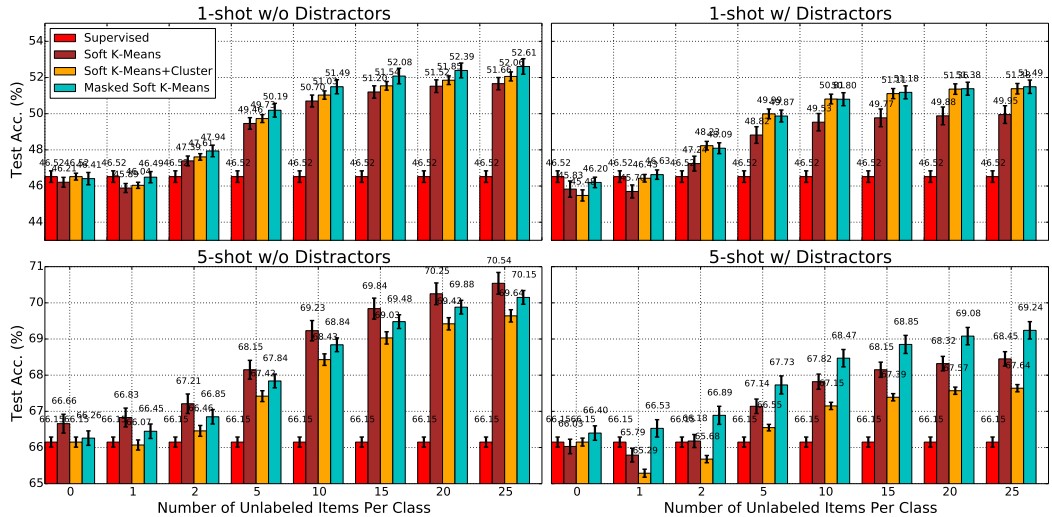

**Figure 6:** Model Performance on *tiered*ImageNet with different number of unlabeled items during test time. We include test accuracy numbers in this chart.

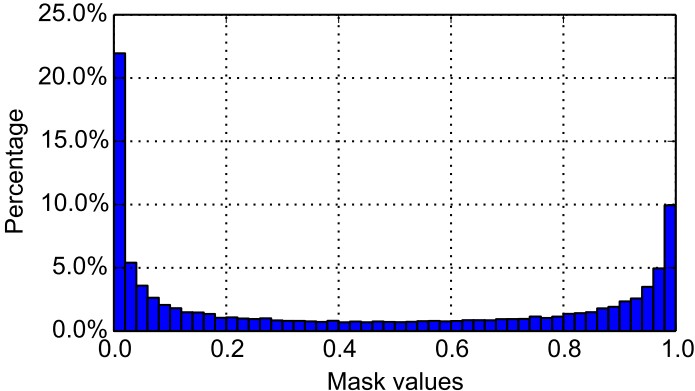

**Figure 7:** Mask values predicted by masked soft k-means on Omniglot.

