# OpenReview forum: "Meta-Learning for Semi-Supervised Few-Shot Classification"
_ICLR.cc/2018/Conference — Accept (Poster)_

### Official Review · AnonReviewer3 · 2017-11-25
**limited novelty**

**Rating:** 6
**Confidence:** 5

**Review:**

This paper is an extension of the “prototypical network” which will be published in NIPS 2017. The classical few-shot learning has been limited to using the unlabeled data, while this paper considers employing the unlabeled examples available to help train each episode. The paper solves a new semi-supervised situation, which is more close to the setting of the real world, with an extension of the prototype network.  Sufficient implementation detail and analysis on results.

However, this is definitely not the first work on semi-supervised formed few-shot learning. There are plenty of works on this topic [R1, R2, R3]. The authors are advised to do a thorough survey of the relevant works in Multimedia and computer vision community.

Another concern is that the novelty. This work is highly incremental since it is an extension of existing prototypical networks by adding the way of leveraging the unlabeled data.

The experiments are also not enough. Not only some other works such as [R1, R2, R3]; but also the other naïve baselines should also be compared, such as directly nearest neighbor classifier, logistic regression, and neural network in traditional supervised learning. Additionally, in the 5-shot non-distractor setting on tiered ImageNet, only the soft kmeans method gets a little bit advantage against the semi-supervised baseline, does it mean that these methods are not always powerful under different dataset?

[R1] “Videostory: A new multimedia embedding for few-example recognition and translation of events,” in ACM MM, 2014

[R2] “Transductive Multi-View Zero-Shot Learning”, IEEE TPAMI 2015

[R3] “Video2vec embeddings recognize events when examples are scarce,” IEEE TPAMI 2014

---

> ### Author Response · Authors · 2017-12-14
> **Response to reviewer 3**
>
> “There are plenty of works on this topic…”
> We also thank the reviewer for pointing out related zero-shot learning literature and we will study them and add those references to the next version of the paper. Based on our preliminary reading, [1] is a journal version that builds on top of [2], with both papers presenting very similar approaches for the application of event recognition in videos. Transductive Multi-View Zero-Shot Learning [3] uses a similar label propagation procedure as ours. However, while [3] uses standalone deep feature extractors, we show that our semi-supervised prototypical network can be trained completely end-to-end. One of the non-trivial results of our paper is that we show that end-to-end meta-learning significantly improves the performance (see Semi-supervised Inference vs. Soft K-means). We would like to emphasize that end-to-end semi-supervised learning in a meta-learning framework is, to the best of our knowledge, a novel contribution.
>
> “...other naïve baselines should also be compared...”
> The recent literature on few-shot learning has established that meta-learning-based approaches outperform kNN and standard neural network based approaches. For the Omniglot dataset, Mann et al. [4] has previously studied baselines such as KNN either in pixel space or deep features, and feedforward NNs. They found these baselines all lag behind their method by quite a lot, and meanwhile Prototypical Networks outperform Mann et al. by another significant margin. For example, Table 1 summarizes the performance for 5-shot, 5-way classification. Therefore, we will provide supervised nearest neighbor, logistic regression, and neural network baselines for completeness; however, we believe that our work is built on top of state-of-the-art methods, and should beat these simple baselines.
>
> Table 1 - Omniglot dataset baselines
> Method             Accuracy
> KNN pixel          48%
> KNN deep         69%
> Mann et al. [4]   88%
> ProtoNet            99.7%
>
> “...not always powerful under different dataset?”
> For completeness we ran both 1-shot and 5-shot settings and found that our method consistently outperforms the baselines. While in 5-shot the improvement is less, this is reasonable since the number of labeled items is larger and the benefit brought by unlabeled items is considerably smaller than in 1-shot settings. We disagree with the comment that our model is not robust under different datasets, since the best settings we found is consistent across all three, quite diverse, datasets, including the novel and much larger tieredImageNet.
>
> References:
> [1] “Video2vec embeddings recognize events when examples are scarce,” IEEE TPAMI 2014
> [2] “Videostory: A new multimedia embedding for few-example recognition and translation of events,” in ACM MM, 2014.
> [3]: Transductive Multi-View Zero-Shot Learning, IEEE TPAMI 2015.
> [4]: One-shot learning with Memory-Augmented Neural Networks. ICML 2016.

---

### Official Review · AnonReviewer1 · 2017-11-25
**The studied problem is interesting, and the paper is well-written. While the proposed method is a natural extension of the existing works.**

**Rating:** 6
**Confidence:** 4

**Review:**

In this paper, the authors studied the problem of semi-supervised few-shot classification, by extending the prototypical networks into the setting of semi-supervised learning with examples from distractor classes.  The studied problem is interesting, and the paper is well-written. Extensive experiments are performed to demonstrate the effectiveness of the proposed methods.  While the proposed method is a natural extension of the existing works (i.e., soft k-means and meta-learning).On top of that, It seems the authors have over-claimed their model capability at the first place as the proposed model cannot properly classify the distractor examples but just only consider them as a single class of outliers. Overall, I would like to vote for a weakly acceptance regarding this paper.

---

> ### Author Response · Authors · 2017-12-14
> **Response to reviewer 1**
>
> Thank you for the comments. We’d like to clarify our setup here: The problem as we have defined it is to correctly perform the given N-way classification in each episode (similarly as in the previous work). Distractors are introduced to make the problem harder in a more realistic way, but the goal is not to be able to classify them. Specifically, our model needs to understand which points are irrelevant for the given classification task (“distractors”) in order to not take them into account, but actually classifying these distractors into separate categories is not required in order to perform the given classification task, so our models make no effort to do this.
>
> Further, we would like to emphasize that adding distractor examples in few-shot classification settings is a novel and more realistic learning environment compared to previous approaches in supervised few-shot learning and as well as concurrent approaches in semi-supervised few-shot learning [1,2]. It is non-trivial to show that various versions of semi-supervised clustering can be trained end-to-end from scratch as another layer on top of prototypical networks, with the presence of distractor clusters  (note that each distractor class has the same number of images as a non-distractor class).
>
> References:
> [1]: Few-Shot Learning with Graph Neural Networks. Anonymous. Submitted to ICLR, 2017.
> [2]: Semi-Supervised Few-Shot Learning with Prototypical Networks. Rinu Boney and Alexander Ilin. CoRR, abs/1711.10856, 2017.

---

### Official Review · AnonReviewer2 · 2017-11-27
**extension of the Prototypical Network to semi-supervised setting**

**Rating:** 6
**Confidence:** 4

**Review:**

This paper proposes to extend the Prototypical Network (NIPS17) to the semi-supervised setting with three possible
strategies. One consists in self-labeling the unlabeled data and then updating the prototypes on the basis of the
assigned pseudo-labels. Another is able to deal with the case of distractors i.e.  unlabeled samples not beloning to
any of the known categories. In practice this second solution is analogous to the first, but a general 'distractor' class
is added. Finally the third technique learns to weight the samples according to their distance to the original prototypes.

These strategies are evaluated in a particular semi-supervised transfer learning setting:  the models are first trained
on some source categories with few labeled data and large unlabeled samples (this setting is derived by subselecting
multiple times a large dataset), then they are used on a final target task with again few labeled data and large
unlabeled samples but beloning to a different set of categories.

+ the paper is well written, well organized and overall easy to read
+/-  this work builds largely on previous work. It introduces only some small technical novelty inspired by soft-k-means
clustering that anyway seems to be effective.
+ different aspect of the problem are analyzed by varying the number of disctractors and varying the level of
semantic relatedness between the source and the target sets

Few notes and questions
1) why for the omniglot experiment the table reports the error results? It would be better to present accuracy as for the other tables/experiments
2) I would suggest to use source and target instead of train and test -- these two last terms are confusing because
actually there is a training phase also at test time.
3) although the paper indicate that there are different other few-shot methods that could be applicable here,
no other approach is considered besides the prothotipical network and its variants. An further external reference
could be used to give an idea of what would be the experimental result at least in the supervised case.

---

> ### Author Response · Authors · 2017-12-14
> **Response to reviewer 2**
>
> We appreciate the constructive comments from reviewer 2 and we are delighted to learn that the reviewer feels that our paper is well written and organized.
>
> “builds largely on previous work… only some small technical novelty…”
> We would like to emphasize that we introduce a new task for few-shot classification, incorporating unlabeled items. This is impactful as follow-up work can use our dataset as a public benchmark. In fact, there are several concurrent ICLR submissions and arxiv pre-prints [1,2] that also introduce semi-supervised few-shot learning. However compared to these concurrent papers, our benchmark extends beyond this work into more realistic and generic settings, with hierarchical class splits and unlabeled distractor classes, which we believe will make positive contributions to the community.
>
> The fact that our semi-supervised prototypical network can be trained end-to-end from scratch is non-trivial, especially under many distractor clusters (note that each distractor class has the same number of images as a non-distractor class). We argue that our extension is simple yet effective, serving as another layer on top of the regular prototypical network layer, and provides consistent improvement in the presence of unlabeled examples. Further, to our knowledge, our best-performing method, the masked soft k-means, is novel.
>
> “It would be better to present accuracy…”
> Thank you for the suggestion. We will revise it in our next version.
>
> “no other approach is considered besides the prototypical network and its variants.”
> ProtoNets is one of the top performing methods for few-shot learning and our proposed extensions each naturally forms another layer on top of the Prototypical layer. To address the concern, we are currently running other variants of the models such as a nearest neighbor baseline, and will report results before the ICLR discussion period ends. In the Omniglot dataset literature, many simple baselines has been extensively explored, and Prototypical Networks are so far the state-of-the-art. Table 1 summarizes the performance for a 5-way 5-shot benchmark (results reported by [3])
>
> Table 1 - Omniglot dataset baselines
> Method             Accuracy
> KNN pixel          48%
> KNN deep         69%
> Mann et al. [3]   88%
> ProtoNet            99.7%
>
> References:
> [1]: Few-Shot Learning with Graph Neural Networks. Anonymous. Submitted to ICLR, 2017.
> [2]: Semi-Supervised Few-Shot Learning with Prototypical Networks. Rinu Boney and Alexander Ilin. CoRR, abs/1711.10856, 2017.
> [3]: One-shot learning with Memory-Augmented Neural Networks. ICML 2016.

---

### Public Comment · (anonymous) · 2017-12-03
**release tiered-imagenet split?**

Great work! Could you release the split for tiered-Imagenet?

---

### Decision · Program_Chairs · 2018-01-29
**ICLR 2018 Conference Acceptance Decision**

**Decision:**

Accept (Poster)

**Comment:**

The paper extends the earlier work on Prototypical networks to semi-supervised setting. Reviewers largely agree that the paper is well-written. There are some concerns on the incremental nature of the paper wrt to the novelty aspect but in the light of reported empirical results which show clear improvement over earlier work and given the importance of the topic, I recommend acceptance.